# Large-scale probabilistic predictors with and without guarantees of validity

**Vladimir Vovk**[*]**, Ivan Petej**[*]**, and Valentina Fedorova**[†]
[*]Department of Computer Science, Royal Holloway, University of London, UK
[†]Yandex, Moscow, Russia
{volodya.vovk,ivan.petej,alushaf}@gmail.com

## Abstract

This paper studies theoretically and empirically a method of turning machine-learning algorithms into probabilistic predictors that automatically enjoys a property of validity (perfect calibration) and is computationally efficient. The price to pay for perfect calibration is that these probabilistic predictors produce imprecise (in practice, almost precise for large data sets) probabilities. When these imprecise probabilities are merged into precise probabilities, the resulting predictors, while losing the theoretical property of perfect calibration, are consistently more accurate than the existing methods in empirical studies.

## 1 Introduction

Prediction algorithms studied in this paper belong to the class of Venn–Abers predictors, introduced in [1]. They are based on the method of isotonic regression [2] and prompted by the observation that when applied in machine learning the method of isotonic regression often produces miscalibrated probability predictions (see, e.g., [3, 4]); it has also been reported ([5], Section 1) that isotonic regression is more prone to overfitting than Platt's scaling [6] when data is scarce. The advantage of Venn–Abers predictors is that they are a special case of Venn predictors ([7], Chapter 6), and so ([7], Theorem 6.6) are always well-calibrated (cf. Proposition 1 below). They can be considered to be a regularized version of the procedure used by [8], which helps them resist overfitting.

The main desiderata for Venn (and related conformal, [7], Chapter 2) predictors are validity, predictive efficiency, and computational efficiency. This paper introduces two computationally efficient versions of Venn–Abers predictors, which we refer to as inductive Venn–Abers predictors (IVAPs) and cross-Venn–Abers predictors (CVAPs). The ways in which they achieve the three desiderata are:

- Validity (in the form of perfect calibration) is satisfied by IVAPs automatically, and the experimental results reported in this paper suggest that it is inherited by CVAPs.

- Predictive efficiency is determined by the predictive efficiency of the underlying learning algorithms (so that the full arsenal of methods of modern machine learning can be brought to bear on the prediction problem at hand).

- Computational efficiency is, again, determined by the computational efficiency of the underlying algorithm; the computational overhead of extracting probabilistic predictions consists of sorting (which takes time $O(n \log n)$, where $n$ is the number of observations) and other computations taking time $O(n)$.

An advantage of Venn prediction over conformal prediction, which also enjoys validity guarantees, is that Venn predictors output probabilities rather than p-values, and probabilities, in the spirit of Bayesian decision theory, can be easily combined with utilities to produce optimal decisions.

In Sections 2 and 3 we discuss IVAPs and CVAPs, respectively. Section 4 is devoted to mini-max ways of merging imprecise probabilities into precise probabilities and thus making IVAPs and CVAPs precise probabilistic predictors.

In this paper we concentrate on binary classification problems, in which the objects to be classified are labelled as 0 or 1. Most of machine learning algorithms are *scoring algorithms*, in that they output a real-valued score for each test object, which is then compared with a threshold to arrive at a categorical prediction, 0 or 1. As precise probabilistic predictors, IVAPs and CVAPs are ways of converting the scores for test objects into numbers in the range $[0, 1]$ that can serve as probabilities, or *calibrating* the scores. In Section 5 we briefly discuss two existing calibration methods, Platt's [6] and the method [8] based on isotonic regression. Section 6 is devoted to experimental comparisons and shows that CVAPs consistently outperform the two existing methods (more extensive experimental studies can be found in [9]).

## 2 Inductive Venn–Abers predictors (IVAPs)

In this paper we consider data sequences (usually loosely referred to as sets) consisting of *observations* $z = (x, y)$, each observation consisting of an *object* $x$ and a *label* $y \in \{0, 1\}$; we only consider binary labels. We are given a training set whose size will be denoted $l$.

This section introduces inductive Venn–Abers predictors. Our main concern is how to implement them efficiently, but as functions, an IVAP is defined in terms of a scoring algorithm (see the last paragraph of the previous section) as follows:

- Divide the training set of size $l$ into two subsets, the *proper training set* of size $m$ and the *calibration set* of size $k$, so that $l = m + k$.
- Train the scoring algorithm on the proper training set.
- Find the scores $s_1, \ldots, s_k$ of the calibration objects $x_1, \ldots, x_k$.
- When a new test object $x$ arrives, compute its score $s$. Fit isotonic regression to $(s_1, y_1), \ldots, (s_k, y_k), (s, 0)$ obtaining a function $f_0$. Fit isotonic regression to $(s_1, y_1), \ldots, (s_k, y_k), (s, 1)$ obtaining a function $f_1$. The multiprobability prediction for the label $y$ of $x$ is the pair $(p_0, p_1) := (f_0(s), f_1(s))$ (intuitively, the prediction is that the probability that $y = 1$ is either $f_0(s)$ or $f_1(s)$).

Notice that the multiprobability prediction $(p_0, p_1)$ output by an IVAP always satisfies $p_0 < p_1$, and so $p_0$ and $p_1$ can be interpreted as the lower and upper probabilities, respectively; in practice, they are close to each other for large training sets.

First we state formally the property of validity of IVAPs (adapting the approach of [1] to IVAPs). A random variable $P$ taking values in $[0, 1]$ is *perfectly calibrated* (as a predictor) for a random variable $Y$ taking values in $\{0, 1\}$ if $\mathbb{E}(Y \mid P) = P$ a.s. A *selector* is a random variable taking values in $\{0, 1\}$. As a general rule, in this paper random variables are denoted by capital letters (e.g., $X$ are random objects and $Y$ are random labels).

**Proposition 1.** *Let $(P_0, P_1)$ be an IVAP's prediction for $X$ based on a training sequence $(X_1, Y_1), \ldots, (X_l, Y_l)$. There is a selector $S$ such that $P_S$ is perfectly calibrated for $Y$ provided the random observations $(X_1, Y_1), \ldots, (X_l, Y_l), (X, Y)$ are i.i.d.*

Our next proposition concerns the computational efficiency of IVAPs; Proposition 1 will be proved later in this section while Proposition 2 is proved in [9].

**Proposition 2.** *Given the scores $s_1, \ldots, s_k$ of the calibration objects, the prediction rule for computing the IVAP's predictions can be computed in time $O(k \log k)$ and space $O(k)$. Its application to each test object takes time $O(\log k)$. Given the sorted scores of the calibration objects, the prediction rule can be computed in time and space $O(k)$.*

Proofs of both statements rely on the geometric representation of isotonic regression as the slope of the GCM (greatest convex minorant) of the CSD (cumulative sum diagram): see [10], pages 9–13 (especially Theorem 1.1). To make our exposition more self-contained, we define both GCM and CSD below.

First we explain how to fit isotonic regression to $(s_1, y_1), \ldots, (s_k, y_k)$ (without necessarily assuming that $s_i$ are the calibration scores and $y_i$ are the calibration labels, which will be needed to cover the use of isotonic regression in IVAPs). We start from sorting all scores $s_1, \ldots, s_k$ in the increasing order and removing the duplicates. (This is the most computationally expensive step in our calibration procedure, $O(k \log k)$ in the worst case.) Let $k' \leq k$ be the number of distinct elements among $s_1, \ldots, s_k$, i.e., the cardinality of the set $\{s_1, \ldots, s_k\}$. Define $s'_j$, $j = 1, \ldots, k'$, to be the $j$th smallest element of $\{s_1, \ldots, s_k\}$, so that $s'_1 < s'_2 < \cdots < s'_{k'}$. Define $w_j := \left|\{i = 1, \ldots, k : s_i = s'_j\}\right|$ to be the number of times $s'_j$ occurs among $s_1, \ldots, s_k$. Finally, define

$$y'_j := \frac{1}{w_j} \sum_{i = 1, \ldots, k : s_i = s'_j} y_i$$

to be the average label corresponding to $s_i = s'_j$.

The *CSD* of $(s_1, y_1), \ldots, (s_k, y_k)$ is the set of points

$$P_i := \left( \sum_{j=1}^{i} w_j, \sum_{j=1}^{i} y'_j w_j \right), \quad i = 0, 1, \ldots, k';$$

in particular, $P_0 = (0, 0)$. The *GCM* is the greatest convex minorant of the CSD. The value at $s'_i$, $i = 1, \ldots, k'$, of the *isotonic regression* fitted to $(s_1, y_1), \ldots, (s_k, y_k)$ is defined to be the slope of the GCM between $\sum_{j=1}^{i-1} w_j$ and $\sum_{j=1}^{i} w_j$; the values at other $s$ are somewhat arbitrary (namely, the value at $s \in (s'_i, s'_{i+1})$ can be set to anything between the left and right slopes of the GCM at $\sum_{j=1}^{i} w_j$) but are never needed in this paper (unlike in the standard use of isotonic regression in machine learning, [8]): e.g., $f_1(s)$ is the value of the isotonic regression fitted to a sequence that already contains $(s, 1)$.

*Proof of Proposition 1.* Set $S := Y$. The statement of the proposition even holds conditionally on knowing the values of $(X_1, Y_1), \ldots, (X_m, Y_m)$ and the multiset $\langle (X_{m+1}, Y_{m+1}), \ldots, (X_l, Y_l), (X, Y) \rangle$; this knowledge allows us to compute the scores $\langle s_1, \ldots, s_k, s \rangle$ of the calibration objects $X_{m+1}, \ldots, X_l$ and the test object $X$. The only remaining randomness is over the equiprobable permutations of $(X_{m+1}, Y_{m+1}), \ldots, (X_l, Y_l), (X, Y)$; in particular, $(s, Y)$ is drawn randomly from the multiset $\langle (s_1, Y_{m+1}), \ldots, (s_k, Y_l), (s, Y) \rangle$. It remains to notice that, according to the GCM construction, the average label of the calibration and test observations corresponding to a given value of $P_S$ is equal to $P_S$. □

The idea behind computing the pair $(f_0(s), f_1(s))$ efficiently is to pre-compute two vectors $F^0$ and $F^1$ storing $f_0(s)$ and $f_1(s)$, respectively, for all possible values of $s$. Let $k'$ and $s'_i$ be as defined above in the case where $s_1, \ldots, s_k$ are the calibration scores and $y_1, \ldots, y_k$ are the corresponding labels. The vectors $F^0$ and $F^1$ are of length $k'$, and for all $i = 1, \ldots, k'$ and both $\epsilon \in \{0, 1\}$, $F_i^\epsilon$ is the value of $f_\epsilon(s)$ when $s = s'_i$. Therefore, for all $i = 1, \ldots, k'$:

- $F_i^1$ is also the value of $f_1(s)$ when $s$ is just to the left of $s'_i$;
- $F_i^0$ is also the value of $f_0(s)$ when $s$ is just to the right of $s'_i$.

Since $f_0$ and $f_1$ can change their values only at the points $s'_i$, the vectors $F^0$ and $F^1$ uniquely determine the functions $f_0$ and $f_1$, respectively. For details of computing $F^0$ and $F^1$, see [9].

**Remark.** There are several algorithms for performing isotonic regression on a partially, rather than linearly, ordered set: see, e.g., [10], Section 2.3 (although one of the algorithms described in that section, the Minimax Order Algorithm, was later shown to be defective [11, 12]). Therefore, IVAPs (and CVAPs below) can be defined in the situation where scores take values only in a partially ordered set; moreover, Proposition 1 will continue to hold. (For the reader familiar with the notion of Venn predictors we could also add that Venn–Abers predictors will continue to be Venn predictors, which follows from the isotonic regression being the average of the original function over certain equivalence classes.) The importance of partially ordered scores stems from the fact that they enable us to benefit from a possible "synergy" between two or more prediction algorithms

---

**Algorithm 1** $\text{CVAP}(T, x)$    //    cross-Venn–Abers predictor for training set $T$

---
1:  split the training set $T$ into $K$ folds $T_1, \ldots, T_K$
2:  **for** $k \in \{1, \ldots, K\}$
3:      $(p_0^k, p_1^k) := \text{IVAP}(T \setminus T_k, T_k, x)$
4:  **return** $\text{GM}(p_1)/(\text{GM}(1 - p_0) + \text{GM}(p_1))$

## 3    Cross Venn–Abers predictors (CVAPs)

A CVAP is just a combination of $K$ IVAPs, where $K$ is the parameter of the algorithm. It is described as Algorithm 1, where $\text{IVAP}(A, B, x)$ stands for the output of IVAP applied to $A$ as proper training set, $B$ as calibration set, and $x$ as test object, and GM stands for geometric mean (so that $\text{GM}(p_1)$ is the geometric mean of $p_1^1, \ldots, p_1^K$ and $\text{GM}(1 - p_0)$ is the geometric mean of $1 - p_0^1, \ldots, 1 - p_0^K$). The folds should be of approximately equal size, and usually the training set is split into folds at random (although we choose contiguous folds in Section 6 to facilitate reproducibility). One way to obtain a random assignment of the training observations to folds (see line 1) is to start from a regular array in which the first $l_1$ observations are assigned to fold 1, the following $l_2$ observations are assigned to fold 2, up to the last $l_K$ observations which are assigned to fold $K$, where $|l_k - l/K| < 1$ for all $k$, and then to apply a random permutation. Remember that the procedure RANDOMIZE-IN-PLACE ([14], Section 5.3) can do the last step in time $O(l)$. See the next section for a justification of the expression $\text{GM}(p_1)/(\text{GM}(1 - p_0) + \text{GM}(p_1))$ used for merging the IVAPs' outputs.

## 4    Making probability predictions out of multiprobability ones

In CVAP (Algorithm 1) we merge the $K$ multiprobability predictions output by $K$ IVAPs. In this section we design a minimax way for merging them, essentially following [1]. For the log-loss function the result is especially simple, $\text{GM}(p_1)/(\text{GM}(1 - p_0) + \text{GM}(p_1))$.

Let us check that $\text{GM}(p_1)/(\text{GM}(1 - p_0) + \text{GM}(p_1))$ is indeed the minimax expression under log loss. Suppose the pairs of lower and upper probabilities to be merged are $(p_0^1, p_1^1), \ldots, (p_0^K, p_1^K)$ and the merged probability is $p$. The extra cumulative loss suffered by $p$ over the correct members $p_1^1, \ldots, p_1^K$ of the pairs when the true label is 1 is

$$\log \frac{p_1^1}{p} + \cdots + \log \frac{p_1^K}{p}, \tag{1}$$

and the extra cumulative loss of $p$ over the correct members of the pairs when the true label is 0 is

$$\log \frac{1 - p_0^1}{1 - p} + \cdots + \log \frac{1 - p_0^K}{1 - p}. \tag{2}$$

Equalizing the two expressions we obtain

$$\frac{p_1^1 \cdots p_1^K}{p^K} = \frac{(1 - p_0^1) \cdots (1 - p_0^K)}{(1 - p)^K},$$

which gives the required minimax expression for the merged probability (since (1) is decreasing and (2) is increasing in $p$).

For the computations in the case of the Brier loss function, see [9].

Notice that the argument above ("conditioned" on the proper training set) is also applicable to IVAP, in which case we need to set $K := 1$; the probability predictor obtained from an IVAP by replacing $(p_0, p_1)$ with $p := p_1/(1 - p_0 + p_1)$ will be referred to as the *log-minimax IVAP*. (And CVAP is log-minimax by definition.)

## 5 Comparison with other calibration methods

The two alternative calibration methods that we consider in this paper are Platt's [6] and isotonic regression [8].

### 5.1 Platt's method

Platt's [6] method uses sigmoids to calibrate the scores. Platt uses a regularization procedure ensuring that the predictions of his method are always in the range

$$\left( \frac{1}{k_- + 2}, \frac{k_+ + 1}{k_+ + 2} \right),$$

where $k_-$ is the number of calibration observations labelled 0 and $k_+$ is the number of calibration observations labelled 1. It is interesting that the predictions output by the log-minimax IVAP are in the same range (except that the end-points are now allowed): see [9].

### 5.2 Isotonic regression

There are two standard uses of isotonic regression: we can train the scoring algorithm using what we call a proper training set, and then use the scores of the observations in a disjoint calibration (also called validation) set for calibrating the scores of test objects (as in [5]); alternatively, we can train the scoring algorithm on the full training set and also use the full training set for calibration (it appears that this was done in [8]). In both cases, however, we can expect to get an infinite log loss when the test set becomes large enough (see [9]).

The presence of regularization is an advantage of Platt's method: e.g., it never suffers an infinite loss when using the log loss function. There is no standard method of regularization for isotonic regression, and we do not apply one.

## 6 Empirical studies

The main loss function (cf., e.g., [15]) that we use in our empirical studies is the *log loss*

$$\lambda_{\log}(p, y) := \begin{cases} -\log p & \text{if } y = 1 \\ -\log(1 - p) & \text{if } y = 0, \end{cases} \tag{3}$$

where $\log$ is binary logarithm, $p \in [0, 1]$ is a probability prediction, and $y \in \{0, 1\}$ is the true label. Another popular loss function is the *Brier loss*

$$\lambda_{\mathrm{Br}}(p, y) := 4(y - p)^2. \tag{4}$$

We choose the coefficient 4 in front of $(y - p)^2$ in (4) and the base 2 of the logarithm in (3) in order for the minimax no-information predictor that always predicts $p := 1/2$ to suffer loss 1. An advantage of the Brier loss function is that it still makes it possible to compare the quality of prediction in cases when prediction algorithms (such as isotonic regression) give a categorical but wrong prediction (and so are simply regarded as infinitely bad when using log loss).

The loss of a probability predictor on a test set will be measured by the arithmetic average of the losses it suffers on the test set, namely, by the *mean log loss* (MLL) and the *mean Brier loss* (MBL)

$$\mathrm{MLL} := \frac{1}{n} \sum_{i=1}^{n} \lambda_{\log}(p_i, y_i), \quad \mathrm{MBL} := \frac{1}{n} \sum_{i=1}^{n} \lambda_{\mathrm{Br}}(p_i, y_i), \tag{5}$$

where $y_i$ are the test labels and $p_i$ are the probability predictions for them. We will not be checking directly whether various calibration methods produce well-calibrated predictions, since it is well

known that lack of calibration increases the loss as measured by loss functions such as log loss and Brier loss (see, e.g., [16] for the most standard decomposition of the latter into the sum of the calibration error and refinement error).

In this section we compare log-minimax IVAPs (i.e., IVAPs whose outputs are replaced by probability predictions, as explained in Section 4) and CVAPs with Platt's method [6] and the standard method [8] based on isotonic regression; the latter two will be referred to as "Platt" and "Isotonic" in our tables and figures. For both IVAPs and CVAPs we use the log-minimax procedure (the Brier-minimax procedure leads to virtually identical empirical results). We use the same underlying algorithms as in [1], namely J48 decision trees (abbreviated to "J48"), J48 decision trees with bagging ("J48 bagging"), logistic regression (sometimes abbreviated to "logistic"), naïve Bayes, neural networks, and support vector machines (SVM), as implemented in Weka [17] (University of Waikato, New Zealand). The underlying algorithms (except for SVM) produce scores in the interval $[0, 1]$, which can be used directly as probability predictions (referred to as "Underlying" in our tables and figures) or can be calibrated using the methods of [6, 8] or the methods proposed in this paper ("IVAP" or "CVAP" in the tables and figures).

For illustrating our results in this paper we use the `adult` data set available from the UCI repository [18] (this is the main data set used in [6] and one of the data sets used in [8]); however, the picture that we observe is typical for other data sets as well (cf. [9]). We use the original split of the data set into a training set of $N_{\text{train}} = 32,561$ observations and a test set of $N_{\text{test}} = 16,281$ observations. The results of applying the four calibration methods (including the vacuous one, corresponding to just using the underlying algorithm) to the six underlying algorithms for this data set are shown in Figure 1. The six top plots report results for the log loss (namely, MLL, as defined in (5)) and the six bottom plots for the Brier loss (namely, MBL). The underlying algorithms are given in the titles of the plots and the calibration methods are represented by different line styles, as explained in the legends. The horizontal axis is labelled by the ratio of the size of the proper training set to that of the calibration set (except for the label `all`, which will be explained later); in particular, in the case of CVAPs it is labelled by the number $K - 1$ one less than the number of folds. In the case of CVAPs, the training set is split into $K$ equal (or as close to being equal as possible) contiguous folds: the first $\lceil N_{\text{train}}/K \rceil$ training observations are included in the first fold, the next $\lceil N_{\text{train}}/K \rceil$ (or $\lfloor N_{\text{train}}/K \rfloor$) in the second fold, etc. (first $\lceil \cdot \rceil$ and then $\lfloor \cdot \rfloor$ is used unless $N_{\text{train}}$ is divisible by $K$). In the case of the other calibration methods, we used the first $\lceil \frac{K-1}{K} N_{\text{train}} \rceil$ training observation as the proper training set (used for training the scoring algorithm) and the rest of the training observations are used as the calibration set.

In the case of log loss, isotonic regression often suffers infinite losses, which is indicated by the absence of the round marker for isotonic regression; e.g., all the log losses for SVM are infinite. We are not trying to use ad hoc solutions, such as clipping predictions to the interval $[\epsilon, 1 - \epsilon]$ for a small $\epsilon > 0$, since we are also using the bounded Brier loss function. The CVAP lines tend to be at the bottom in all plots; experiments with other data sets also confirm this.

The column `all` in the plots of Figure 1 refers to using the full training set as both the proper training set and calibration set. (In our official definition of IVAP we require that the last two sets be disjoint, but in this section we continue to refer to IVAPs modified in this way simply as IVAPs; in [1], such prediction algorithms were referred to as SVAPs, simplified Venn–Abers predictors.) Using the full training set as both the proper training set and calibration set might appear naïve (and is never used in the extensive empirical study [5]), but it often leads to good empirical results on larger data sets. However, it can also lead to very poor results, as in the case of "J48 bagging" (for IVAP, Platt, and Isotonic), the underlying algorithm that achieves the best performance in Figure 1.

A natural question is whether CVAPs perform better than the alternative calibration methods in Figure 1 (and our other experiments) because of applying cross-over (in moving from IVAP to CVAP) or because of the extra regularization used in IVAPs. The first reason is undoubtedly important for both loss functions and the second for the log loss function. The second reason plays a smaller role for Brier loss for relatively large data sets (in the lower half of Figure 1 the curve for `Isotonic` and `IVAP` are very close to each other), but IVAPs are consistently better for smaller data sets even when using Brier loss. In Tables 1 and 2 we apply the four calibration methods and six underlying algorithms to a much smaller training set, namely to the first $5,000$ observations of the `adult` data set as the new training set, following [5]; the first $4,000$ training observations are used as the proper training set, the following $1,000$ training observations as the calibration set, and all other observa-

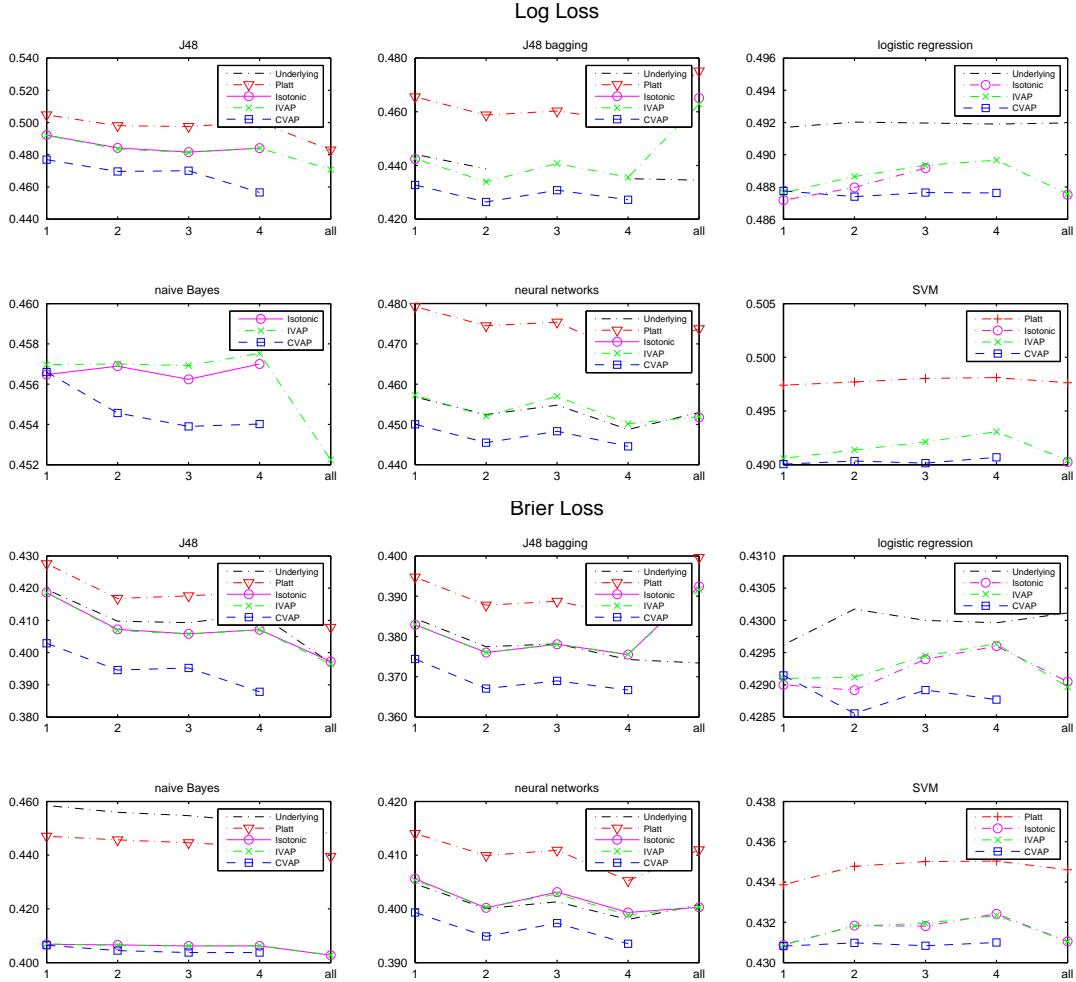

Figure 1: The log and Brier losses of the four calibration methods applied to the six prediction algorithms on the `adult` data set. The numbers on the horizontal axis are ratios $m/k$ of the size of the proper training set to the size of the calibration set; in the case of CVAPs, they can also be expressed as $K-1$, where $K$ is the number of folds (therefore, column 4 corresponds to the standard choice of 5 folds in the method of cross-validation). Missing curves or points on curves mean that the corresponding values either are too big and would squeeze unacceptably the interesting parts of the plot if shown or are infinite (such as many results for isotonic regression and J48 for log loss).

tions (the remaining training and all test observations) are used as the new test set. The results are shown in Tables 1 for log loss and 2 for Brier loss. They are consistently better for IVAP than for IR (isotonic regression). Results for nine very small data sets are given in Tables 1 and 2 of [1], where the results for IVAP (with the full training set used as both proper training and calibration sets, labelled "SVA" in the tables in [1]) are consistently (in 52 cases out of the 54 using Brier loss) better, usually significantly better, than for isotonic regression (referred to as DIR in the tables in [1]).

The following information might help the reader in reproducing our results (in addition to our code being publicly available [9]). For each of the standard prediction algorithms within Weka that we use, we optimise the parameters by minimising the Brier loss on the calibration set, apart from the column labelled `all`. (We cannot use the log loss since it is often infinite in the case of isotonic regression.) We then use the trained algorithm to generate the scores for the calibration and test sets, which allows us to compute probability predictions using Platt's method, isotonic regression, IVAP,

Table 1: The log loss for the four calibration methods and six underlying algorithms for a small subset of the `adult` data set

| algorithm | Platt | IR | IVAP | CVAP |
|---|---|---|---|---|
| J48 | 0.5226 | $\infty$ | 0.5117 | 0.5102 |
| J48 bagging | 0.4949 | $\infty$ | 0.4733 | 0.4602 |
| logistic | 0.5111 | $\infty$ | 0.4981 | 0.4948 |
| naïve Bayes | 0.5534 | $\infty$ | 0.4839 | 0.4747 |
| neural networks | 0.5175 | $\infty$ | 0.5023 | 0.4805 |
| SVM | 0.5221 | $\infty$ | 0.5015 | 0.4997 |

Table 2: The analogue of Table 1 for the Brier loss

| algorithm | Platt | IR | IVAP | CVAP |
|---|---|---|---|---|
| J48 | 0.4463 | 0.4378 | 0.4370 | 0.4368 |
| J48 bagging | 0.4225 | 0.4153 | 0.4123 | 0.3990 |
| logistic | 0.4470 | 0.4417 | 0.4377 | 0.4342 |
| naïve Bayes | 0.4670 | 0.4329 | 0.4311 | 0.4227 |
| neural networks | 0.4525 | 0.4574 | 0.4440 | 0.4234 |
| SVM | 0.4550 | 0.4450 | 0.4408 | 0.4375 |

and CVAP. All the scores apart from SVM are already in the $[0, 1]$ range and can be used as probability predictions. Most of the parameters are set to their default values, and the only parameters that are optimised are `C` (pruning confidence) for J48 and J48 bagging, `R` (ridge) for logistic regression, `L` (learning rate) and `M` (momentum) for neural networks (`MultilayerPerceptron`), and `C` (complexity constant) for SVM (`SMO`, with the linear kernel); naïve Bayes does not involve any parameters. Notice that none of these parameters are "hyperparameters", in that they do not control the flexibility of the fitted prediction rule directly; this allows us to optimize the parameters on the training set for the `all` column. In the case of CVAPs, we optimise the parameters by minimising the cumulative Brier loss over all folds (so that the same parameters are used for all folds). To apply Platt's method to calibrate the scores generated by the underlying algorithms we use logistic regression, namely the function `mnrfit` within MATLAB's Statistics toolbox. For isotonic regression calibration we use the implementation of the PAVA in the R package `fdrtool` (namely, the function `monoreg`).

For further experimental results, see [9].

# 7 Conclusion

This paper introduces two new algorithms for probabilistic prediction, IVAP, which can be regarded as a regularised form of the calibration method based on isotonic regression, and CVAP, which is built on top of IVAP using the idea of cross-validation. Whereas IVAPs are automatically perfectly calibrated, the advantage of CVAPs is in their good empirical performance.

This paper does not study empirically upper and lower probabilities produced by IVAPs and CVAPs, whereas the distance between them provides information about the reliability of the merged probability prediction. Finding interesting ways of using this extra information is one of the directions of further research.

**Acknowledgments**

We are grateful to the conference reviewers for numerous helpful comments and observations, to Vladimir Vapnik for sharing his ideas about exploiting synergy between different learning algorithms, and to participants of the conference *Machine Learning: Prospects and Applications* (October 2015, Berlin) for their questions and comments. The first author has been partially supported by EPSRC (grant EP/K033344/1) and AFOSR (grant "Semantic Completions"). The second and third authors are grateful to their home institutions for funding their trips to Montréal.

## Footnotes

[13]. Suppose, e.g., that one prediction algorithm outputs (scalar) scores $s_1^1, \ldots, s_k^1$ for the calibration objects $x_1, \ldots, x_k$ and another outputs $s_1^2, \ldots, s_k^2$ for the same calibration objects; we would like to use both sets of scores. We could merge the two sets of scores into composite vector scores, $s_i := (s_i^1, s_i^2), i = 1, \ldots, k$, and then classify a new object $x$ as described earlier using its composite score $s := (s^1, s^2)$, where $s^1$ and $s^2$ are the scalar scores computed by the two algorithms. Preliminary results reported in [13] in a related context suggest that the resulting predictor can outperform predictors based on the individual scalar scores. However, we will not pursue this idea further in this paper.

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
