[Reviews · NeurIPS 2015]

Submitted by Assigned_Reviewer_1

This paper tackles the very significant problem of converting arbitrary class membership scores into well-calibrated probabilities.

It appears to provide a very useful technique that works well.

The paper covers both theory and algorithmic issues.

It is well written and easy to follow.
Summary: Extends isotonic regression by using cross validation to generate multiple scoring sets and combining the scores generated on each.

Submitted by Assigned_Reviewer_2

The paper proposes two methods for turning scoring outputs into probabilities as alternatives to methods such as Platt's method and isotonic regression. It is clear and well written, and the authores are in particular applauded for giving information relevant to reproducing the experimental results. The methods proposed are original, and this is a sufficiently widespread problem that any demonstrably improved method is a significant advance. The paper is therefore very suitable for NIPS. I have some reservation regarding the experimental results in that, whereas in figure 1 CVAP seems to attain a clear advantage, the numbers presented in tables 1 and 2 are close enough to be less convincing --- can the authors provide any further justification here?
Summary: A nice paper, suitable for NIPS.

Submitted by Assigned_Reviewer_3

This paper proposes two techniques for post-processing a score-valued output of a binary classifier by converting it into an estimate of a class probability.

The techniques are simple and efficient. It is surely publishable in principle, however, in my perspective the paper in its current form has far too many weak spots to be acceptable. They are listed in the following.

Until the conclusion the distinction between contribution and background remains fuzzy.

The content of section 2 seems trivial, in particular proposition 2. All it takes is sorting and binary search. However, this seems to be one of the "major" contributions.

The hyperparameters of the learning machines were set by minimizing the training error?! E.g., a Gaussian kernel SVMs can easily achieve zero error. Then all we see is over-fitting. This is a clear no-go, it renders all experimental results meaningless. That alone is a reason for rejection.

The improvement over the baseline methods is consistent, but the effect size is small. So are the new methods relevant, in particular since they take more space and time than Platt's method? I am missing a discussion of this point. For acceptance I want to be convinced that somebody out there cares.

Minor comments:

In contrast to what it stated in the paper, Platt's method is not invariant w.r.t. the (sigmoidal?) conversion of the scores to the unit interval. It does not become 100% clear whether the conversion is actually performed with Platt's method or not.

At some point the paper mentions that most models output values in the unit interval, which could hence be interpreted as probabilities. This point is not followed upon in the experimental evaluation. Why not?

In section 5.2 it is claimed that there is no simple ad-hoc regularization for isotonic regression. I don't see why Platt's regularization technique cannot be applied, which basically amounts to adding two virtual scores of plus and minus infinity.

The first word of the title reads "Large-scale". Actually, nothing in this paper is large scale, and I don't see any relation to processing of large-scale data.

The plots in figure 1 are far too small to be readable in a standard b/w printout.
Summary: This paper has too many weak spots to be publishable.

Submitted by Assigned_Reviewer_4

This paper proposed two algorithms for probabilistic predictions, IVAP and CVAP. At the very beginning, I need to say I am not an expert on this topic. But as for me, the idea in this paper is interesting. The methods are easy to understand and use. Theoretical results are provided to show the validity of the proposed IVAP. Experimental results justify the performance of the proposed methods.

My first concern is how this proposed method is different from and better than the Venn-Abers predictors. The authors should add some more introduction about the Venn-Abers predictors by showing its validity,predictive efficiency and computational efficiency first, then make the comparison between the original algorithm and the proposed ones.

My second concern is that the experiment used only one UCI dataset to justify the performance, which may not be very solid. The authors should try on more datasets and more challenging datasets other than UCI datasets to better confirm the superiority of the proposed method. Also, as CVAP generally obtains the best performance. The authors should try to show the sensitivity results by vary different k used in CVAP.

If the authors can well address the above concerns in the rebuttal, I would like to accept this paper.

I have read the author feedback. I am happy to see some more datasets reported in the supplementary. But it is still not clearly clarified why the original Venn-Aber is not feasible on the current used dataset with this scale. So I urge the authors to reorganize the content in the introduction and move some more experimental results back into the main text.
Summary: Overall, this paper shows some interesting ideas and promising results. But the novelty of this paper needs to be clarified and the experimental design needs to be improved.

Author Feedback
Author rebuttal: We are grateful to the reviewers for their effort. There are some repetitions below as we wanted to make our responses to different reviewers self-contained (which we achieved with one exception).

Reviewer 2:

The inclusion of just one dataset was primarily due to the page limit. We include additional results in the supplementary material and show that the results hold across a variety of datasets.

The numbers in the table refer to a smaller training set than the original UCI adult dataset (i.e., 5000, vs. 32156) as well as a smaller calibration set (only 1000 examples). The advantage of IVAP/CVAP over standard calibration methods is still present albeit with a slightly lower advantage than when a larger training/calibration set is used.

Reviewer 3:

The main difference of the proposed algorithms from Venn-Abers predictors is that they are computationally efficient; for example, running Venn-Abers predictors on the adult data set would not be feasible. In the main paper we use only one dataset, but there are more in supplementary material. The sensitivity to different k is shown to an extent in the charts (perhaps we can also say that we have experiments with a greater number of folds, up to 10, and it does not affect the conclusion of CVAP being overwhelmingly more accurate).

Reviewer 4:

No answer is required.

Reviewer 5:

We do not think that it is fair to say that Proposition 2 only involves sorting and binary search. Unfortunately, the details of the algorithms mentioned in Section 2 could only be given in Supplementary Material. Even the simple Algorithm 1 of Supplementary Material is a modification (albeit trivial, as we admit there) of Graham's scan. It appears to us that Algorithm 2 is new.

The number of parameters that we tune is always small, 2 or 3, and none of them controls directly the flexibility of fitting (as is the case with the parameter of Gaussian kernel SVMs; we use just the linear kernel). Therefore, we do not expect significantly different results for alternative ways of fitting parameters. The standard alternative way is cross-validation, but at this time our preferred way is Caruana and Niculescu-Mizil's [5] use of the calibration set for choosing the best values of parameters; a big advantage of this method over our current method is that it can be applied to underlying algorithms with any parameters and hyperparameters (including those that control flexibility directly). We have had time to do comprehensive experiments for Caruana and Niculescu-Mizil's method only for the Bank Marketing data set; interestingly, it gives better results than our method only for CVAPs (but the difference is tiny, and this is very likely to be a coincidence). We will know the results of the comparison for other data sets in a few days and will include them in the next version of the paper.

No, we do not convert the scores to the unit interval before applying Platt's method.

We do not compare the results obtained using calibration methods with the results produced directly by the underlying algorithms (provided the scores are already in the interval [0,1]) since the latter are known to be typically miscalibrated and leading to poor results: see, e.g., [5] and [1]. (We should have mentioned this explicitly, and will do so in the next version of the paper.)

Platt has a justification for his method; but we agree that adding two virtual observations (one with score \infty and label 0 and the other with score -\infty and label 1) is somewhat analogous.

What we mean by "large-scale" is that our methods are large-scale as compared to the previous work on Venn-Abers predictors, which would not be feasible even for a dataset consisting of a total of 50,000 observations; besides, there are no obvious obstacles to combining them with, say, MapReduce techniques and using for really large data sets (especially that processing different folds can be done in parallel).

Reviewer 6:

Our intuition is that by adding the test object (with different postulated labels) to the training set we regularize the method.

Reviewer 7:

The main contribution over [1] is that our new methods are computationally efficient; the methods of [1] have similar theoretical guarantees but would have been infeasible for, e.g., the adult data set.

While cross-validation is used in the LibSVM package to generate probabilistic outputs by default, within the SMO Weka package it is not. We believe that in our experiments there is reduced potential for overfitting as the number of parameters that we tune is always small, 2 or 3, and none of them controls directly the flexibility of fitting. The standard alternative way is cross-validation, but at this time our preferred way is Caruana and Niculescu-Mizil's [5] use of the calibration set for choosing the best values of parameters. See also our response to Reviewer 5.